# The Success and Safety of Endoscopic Retrograde Cholangiopancreatography in Surgically Altered Gastrointestinal Anatomy

**DOI:** 10.3390/medsci13010018

**Published:** 2025-02-11

**Authors:** Samuel Han, Jennifer M. Kolb, Steven A. Edmundowicz, Augustin R. Attwell, Hazem T. Hammad, Sachin Wani, Raj J. Shah

**Affiliations:** 1Division of Gastroenterology and Hepatology, Mayo Clinic, Rochester, MN 55905, USA; samuel.y.han@gmail.com; 2Division of Digestive Diseases, University of California Los Angeles, Los Angeles, CA 90024, USA; jkolb@mednet.ucla.edu; 3Division of Gastroenterology and Hepatology, University of Colorado Anschutz Medical Campus, Aurora, CO 80045, USA; steven.edmundowicz@cuanschutz.edu (S.A.E.); augustin.attwell@dhha.org (A.R.A.); hazem.hammad@cuanschutz.edu (H.T.H.); sachin.wani@cuanschutz.edu (S.W.)

**Keywords:** ERCP, altered anatomy, Whipple, Roux-en-Y gastric bypass, Roux-en-Y hepaticojejunostomy, single-balloon enteroscopy, double-balloon enteroscopy

## Abstract

Background/Objectives: Performing endoscopic retrograde cholangiopancreatography (ERCP) in surgically altered gastrointestinal anatomy remains challenging, frequently necessitating the use of forward-viewing endoscopes. Given the challenge in endoscope selection based on the type of altered anatomy, the aim of this study was to examine ERCP success rates by specific endoscopes for different anatomy types. Methods: This single-center retrospective study examined ERCPs performed in patients with surgically altered gastrointestinal anatomy during an 18-year period. Enteroscopy success, cannulation success, and intervention success rates were compared between the different anatomy and endoscope types. Results: This study included a total of 334 adult patients (665 total ERCPs) with altered anatomy. The pediatric colonoscope was most frequently utilized (32.2%), and the majority of procedures were performed for biliary indications. Enteroscopy success was 82.2% in Roux-en-Y gastric bypass (RYGB), 97% in Billroth II, 91.5% in Whipple, and 93.2% in Roux-en-Y hepaticojejunostomy (RYHJ). Cannulation success was 90.5% in RYGB, 90.5% in Billroth II, 83.6% in Whipple, and 90.6% in RYHJ. Intervention success was 88.2% in Billroth II, 65.1% in RYGB, 81.6% in Whipple, and 87.5% in RYHJ. In patients with RYGB and RYHJ, SBE was utilized most frequently, with rotational enteroscopy having the highest success rates. The overall adverse event rate was 5.1%, with the majority of these being mild in severity. Conclusions: This large retrospective study found ERCP with forward-viewing endoscopes to be safe and effective for a variety of surgically altered anatomy types. Despite recent advances seen with endoscopic ultrasound-guided drainage procedures, this study advocates for ERCP as the initial approach for pancreaticobiliary access in surgically altered anatomy.

## 1. Introduction

Performance of ERCP in the surgically altered upper gastrointestinal anatomy poses a significant challenge for the therapeutic endoscopist, particularly in light of the increase in the number of bariatric surgeries [1,2]. Through emerging technologies, the endoscopist now has a myriad of methods in their toolbox to tackle these difficult cases. Device-assisted ERCP primarily entails double-balloon, single-balloon, and rotational overtube-assisted enteroscopy, where the use of an overtube in a forward-viewing device facilitates afferent limb access [3]. For Roux-Y gastric bypass (RYGB) patients, laparoscopy-assisted ERCP represents one of the most effective methods but mandates surgical assistance in placing a trocar into the excluded stomach and is associated with longer procedure times in addition to laparoscopy-related adverse events, including post-operative infections [4]. Percutaneous transhepatic access may also facilitate biliary intervention but may require numerous procedures and long resolution periods [5]. More recent techniques, such as the EUS-directed transgastric ERCP (EDGE), incorporate therapeutic EUS to enable standard ERCP through a lumen-apposing metal stent placed into the excluded stomach but will typically require multiple sessions to allow for tract maturation and risk of weight regain in the setting of persistent gastro-gastric fistula [6,7].

Despite the increasing number of ERCP techniques available for pancreaticobiliary access in patients with surgically altered anatomy, highly variable rates of success have been reported [8,9]. Looking specifically at device-assisted ERCP for RYGB patients, enteroscopy success (reaching the papilla and/or pancreaticobiliary anastomosis) ranged from 55 to 100% [8]. Breaking this down further into different types of anatomy, pooled enteroscopy success rates range from 80% in Roux-en-Y gastric bypass (RYGB) anatomy to as high as 96% in Billroth II anatomy [10]. Actual success of ERCP, which requires cannulation of the desired duct as well as completion of the desired intervention, was generally lower, with overall success rates of 61.7% to 74% [8,10]. Major adverse events occur in 3.4% to 6.5% of procedures, primarily consisting of post-ERCP pancreatitis, perforation, and bleeding [8,10,11].

In light of the numerous ERCP techniques available, limited data exist regarding the use of different endoscope selections based on the type of surgically altered anatomy. Therefore, the primary aim of this study was to compare ERCP success rates in different types of surgically altered anatomy.

## 2. Materials and Methods

This was a single-center retrospective study examining all ERCPs performed on patients with surgically altered upper gastrointestinal anatomy at a tertiary academic medical center spanning a 17-year period. Surgically altered anatomy types included Billroth II, Roux-en-y hepaticojejunostomy (RYHJ), RYGB, classic pancreaticoduodenectomy (Whipple), and pylorus-preserving pancreaticoduodenectomy (PPPD). Endoscopes used in these procedures included a standard duodenoscope (Olympus, Inc., Center Valley, PA, USA), adult and pediatric colonoscopes (Olympus), single-balloon enteroscope (SBE, SIF Q180, Olympus), double-balloon enteroscope (DBE, EN-450T5, EC-450B15, Fujifilm Endoscopy, Wayne, NJ, USA), and rotational enteroscope (Spirus Medical, Stoughton, MA, USA). All procedures were performed by highly experienced endoscopists (>3000 ERCPs performed each) using standard technique as has previously been described [3,12,13,14,15]. If the index ERCP attempt was unsuccessful, further endoscopic attempts were made at the discretion of the endoscopist. Information collected included patient demographics, procedure indication, type of endoscope, intervention performed, and adverse events. Institutional review board approval (COMIRB 17-1630) was obtained for this study.

### 2.1. Definitions

Enteroscopy success was defined as reaching the papilla and/or pancreaticobiliary anastomosis with the endoscope. Cannulation success was defined as cannulation of the desired duct with successful cholangiogram/pancreatogram. The primary outcome, intervention success, was defined as completion of the desired intervention.

Adverse events (AEs) were classified in accordance with the American Society for Gastrointestinal Endoscopy (ASGE) guidelines for ERCP-related adverse events [16].

### 2.2. Statistical Analysis

Descriptive statistics were used to describe patient demographics and procedural information. The chi-square test or the Fisher exact test was used to compare the intervention success rates between the different anatomies and endoscopes. A *p* value < 0.05 was considered significant. All statistical analysis was performed using Stata 15.1 (StataCorp, College Station, TX, USA).

## 3. Results

### 3.1. Study Patients

A total of 334 adult patients with surgically altered upper gastrointestinal anatomy were included in this study (Figure 1). The mean age was 56.7 (standard deviation SD, 15.5) years, and females accounted for 50% (n = 167) of the study cohort (Table 1). The most common anatomy types included RYGB (n = 100 subjects), followed by the Billroth II (n = 81), Whipple (n = 76), and RYHJ (n = 63) anatomy types. The pediatric colonoscope (32.2%) was used most frequently, followed by the single-balloon enteroscope (31.6%). Indications were predominantly biliary (n = 309, 92.5%), with the most common indications including biliary pain (28.1%), dilated bile duct on imaging (23.7%), elevated liver function tests (20.7%), and biliary stones (19.8%). Sphincterotomy was performed in 29.3% of ERCPs, with dilation performed in 60.8% of ERCPs.

### 3.2. Procedural Outcomes

The cohort received a total of 665 ERCPs (mean of 1.99 procedures/patient) during the study period. The overall enteroscopy, cannulation, and intervention success rates were 91.1% (n = 606), 83.8% (n = 557), and 81.2% (n = 540), respectively. Broken down by anatomy type, enteroscopy success was 82.2% in RYGB, 97% in Billroth II, 91.5% in Whipple, and 93.2% in RYHJ (Table 2). Cannulation success was 90.5% in RYGB, 90.5% in Billroth II, 83.6% in Whipple, and 90.6% in RYHJ. Intervention success was 88.2% in Billroth II, 65.1% in RYGB, 81.6% in Whipple, and 87.5% in RYHJ.

For biliary indications, enteroscopy success, cannulation success, and intervention success rates were 80.6%, 75.8%, and 68.7%, respectively. For pancreatic indications, enteroscopy success, cannulation success, and intervention success rates were 76%, 64%, and 60%, respectively.

### 3.3. Success Rates by Endoscope Type Used for Each Altered Anatomy Classification

Overall, ERCP with a duodenoscope (only in Billroth II anatomy) had the highest enteroscopy, cannulation, and intervention success rates (Table 3). Of the device-assisted enteroscopes, rotational enteroscopy had the highest enteroscopy, cannulation, and intervention success rates. DBE was associated with the lowest enteroscopy, cannulation, and intervention success rates.

A standard duodenoscope was used most frequently in Billroth II anatomy, with a 99.3% enteroscopy success rate, 96.5% cannulation success rate, and 94.3% intervention success rate (Appendix A).

For RYHJ, SBE was most commonly utilized, with a 94.2% enteroscopy success rate, 92.8% cannulation success rate, and 89.1% intervention success rate (Table 4). Among the six participants with initial enteroscopy failure where ERCP was re-attempted, three had successful ERCPs on their second attempt. Repeat ERCP with SBE was successful in three out of four participants where SBE was initially attempted, while repeat ERCP with DBE (index with DBE) and SBE (index with rotational enteroscopy) failed in two participants due to angulation and a long afferent limb.

ERCP in RYGB was most commonly performed with SBE. The enteroscopy success rate was 83.3%, the cannulation success rate was 68.5%, and the intervention success rate was 64.8% (Table 5). Rotational enteroscopy had the highest success rates, with a 100% enteroscopy success rate, 75% cannulation success rate, and 75% intervention success rate. Repeat endoscopy was not attempted in the 26 participants with initial enteroscopy failure.

In classic Whipple’s anatomy, the pediatric colonoscope was used in the majority of cases, with an enteroscopy success rate of 93.1%, cannulation success rate of 85.3%, and intervention success rate of 82.8% (Appendix A). Repeat ERCP was attempted in three participants, with success achieved in one participant who had an SBE-assisted ERCP (initial attempt with SBE). In one case where the initial attempt was made with a pediatric colonoscope, a repeat attempt with SBE failed (excessive looping), and in the remaining case where SBE was utilized initially, a repeat attempt with a pediatric colonoscope failed due to a severe stricture.

For pylorus-preserving Whipple anatomy, ERCP with a pediatric colonoscope had an enteroscopy success rate of 76.2%, a cannulation success rate of 66.7%, and an intervention success rate of 66.7% (Appendix A). The three cases of enteroscopy failure were secondary to obstruction within the afferent limb.

### 3.4. Comparison of Success Rates Between Anatomy Types

There was a significant difference between the various surgically altered upper gastrointestinal anatomy types in enteroscopy success, cannulation success, and intervention success. Enteroscopy success was significantly higher in Billroth II, RYHJ, and Whipple’s anatomy types compared to patients with RYGB and PPPD (*p* < 0.001). Cannulation success was significantly higher in patients with Billroth II and RYHJ anatomies than in other anatomy types (*p* < 0.001). Lastly, intervention success was significantly higher in Billroth II, RYHJ, and classic Whipple anatomy types (*p* < 0.001).

### 3.5. Adverse Events

The overall AE rate was 5.1%, with post-ERCP pancreatitis occurring in 1.2% of cases (Table 1). The majority of AEs were mild in severity (88.2%), with two cases (5.9%, both perforations) being severe. There were four perforations (0.6%), with rotational enteroscopy having the highest rate (3.3%, n = 2). Of these four perforations, two were managed conservatively, while two required surgical repair.

## 4. Discussion

In this large retrospective study, we found performing ERCP without surgical assistance in surgically altered upper gastrointestinal anatomy to be safe and effective. Not surprisingly, ERCP in Billroth II anatomy carried the highest success rates, but we also demonstrated a high success rate in performing ERCP in RYHJ anatomy, particularly with the SBE. Intervention success was lowest in patients with RYGB, highlighting the particular difficulty with this anatomy type, and while currently unavailable in clinical practice in the USA, we also demonstrate the relatively high success rates seen with the rotational enteroscopy system. These results support the consideration of ERCP in patients with surgically altered upper gastrointestinal anatomy prior to percutaneous, EUS-guided, or surgical interventions, particularly given the higher morbidity and greater length of stay seen with alternative approaches [5,9,17].

In Billroth II anatomy, the afferent limb from the duodenum poses a challenge in itself to intubate. An acutely angled afferent limb or a relatively short lesser curvature can create quite a challenge with the duodenoscope in particular [18]. Even upon achieving visualization of the major papilla, the reverse orientation of the papilla makes successful cannulation difficult and also creates a counterintuitive sphincterotomy direction aiming towards the 6 o’clock position. In one of the largest series detailing ERCP in Billroth II anatomy, Bove et al. reported a duodenal intubation (enteroscopy success) rate of 86.7% with a 93.8% cannulation success rate upon successful duodenal intubation [19]. The inability to reach the duodenum in over 15% of cases with the duodenoscope reflects our results, where duodenal intubation required a forward-viewing endoscope, including a balloon-assisted enteroscope in nearly 15% of our ERCPs. This likely reflects the particular difficulty with variably long and angulated afferent loops in this anatomy. Additionally, the risk of peritoneal perforation at the gastrojejunal anastomosis and the ligament of Treitz must be noted, given the pressure exerted by the endoscope at these areas [19].

Bile duct reconstruction with RYHJ anatomy may be performed for iatrogenic bile duct injuries or liver transplantation and rarely for benign biliary strictures that are not responsive to endoscopic therapy. From a post-RYHJ ERCP perspective, a key aspect of the procedure involves intubating the biliary limb and advancing to the biliary orifice upon reaching the jejunojejunostomy. Itokawa et al. described an 85.3% enteroscopy success rate with either standard SBE or DBE but only a 50% success rate using the short double balloon-assisted enteroscopy system [20]. Similarly, Azeem et al. reported a 91.4% enteroscopy success rate, 75.9% cannulation success rate, and an intervention success rate of 75.9% in 58 ERCPs with SBE, finding a significantly higher therapeutic success rate using SBE compared to the pediatric colonoscope [21]. A multicenter study from Europe found the highest success rates with DBE, with an intervention success rate of 86%. While DBE had a relatively low success rate in RYHJ in our study, this may reflect sample bias given the low number of DBE cases and the likely use of DBE in anticipated particularly long or angulated biliary limbs where SBE or rotational enteroscopy had failed in reaching the hepaticojejunostomy.

ERCP in RYGB likely represents the most challenging of the surgically altered anatomies and is now often performed via EDGE procedure or laparoscopy-assisted ERCP in the setting of concomitant cholecystectomy in appropriate patients [22,23]. In a large single-center retrospective study, Ishii et al. reported an enteroscopy success rate of 93.5% using either SBE or DBE with an intervention success rate of 88.1% [24]. As standard cannulation techniques were successful in only 67.8% of cases, advanced cannulation techniques, including precut sphincterotomy (16.3%) and the double-guidewire technique (12.5%), were frequently utilized to facilitate cannulation. In contrast, a multicenter study from the US examining SBE, DBE, and rotational enteroscopy included 63 patients with RYGB, finding an overall enteroscopy success rate of 76% and intervention success rate of 62% with the greatest intervention success rate seen with DBE (67%) [3]. In the present study, we found the highest success rates with rotational enteroscopy, with an enteroscopy success rate of 100% and an intervention success rate of 75%. Cannulating a preserved major papilla and performing therapeutic maneuvers with a forward-viewing endoscope without an elevator represents the primary challenge in ERCP in RYGB, a limitation that Ishii et al. addressed by removing the enteroscope while keeping the overtube in place and cutting a hole in the overtube to allow for insertion of an oblique viewing gastroscope (GIF-XK240, Olympus) or an ultrathin endoscope, although the use of these endoscopes is limited by their small working channel [24]. Our finding of an intervention success rate in nearly two-thirds of patients suggests, however, that device-assisted ERCP can be attempted initially with relatively low risk. Nevertheless, in the context of recent data suggesting enhanced weight loss in patients with longer biliopancreatic limbs, performing device-assisted ERCP may become increasingly more challenging, and careful discussion is warranted with patients (preferably in the outpatient setting) that factors in their specific anatomy and alternative techniques such as the EDGE procedure or laparoscopy-assisted ERCP when the gallbladder is in situ if laparoscopic cholecystectomy is also required [25,26].

In the traditional pancreaticoduodenectomy anatomy, ERCP requires successful intubation of the afferent limb at the gastrojejunostomy. Biliary limbs have variable lengths, and initial experiences with ERCP in this anatomy carried low success rates, particularly for pancreatic interventions [1]. More recent studies have demonstrated a higher intervention success rate when using balloon-assisted enteroscopes compared to standard forward-viewing endoscopes such as a pediatric colonoscope [27]. Our study predominantly utilized the pediatric colonoscope for this anatomy type, finding a relatively high intervention success rate, which is in line with a recent study demonstrating a technical success rate of 90.9% using a pediatric colonoscope in patients with pancreaticoduodenectomy anatomy [28]. Interestingly, overtube-assisted enteroscopy was required for 22 procedures, during which enteroscopy success was still limited to 77.2%, demonstrating the endoscopic challenge, especially of visualizing the biliary or pancreatic anastomoses often due to angulations within the biliary limb and accounting for a large proportion of ERCP failures in post-Whipple patients [1]. Our pylorus-preserving pancreaticoduodenectomy anatomy population was small but represents another particularly challenging cohort in the setting of malignant obstruction of the biliary limb. Additionally, endoscopy can be challenging in this anatomy type due to relatively long biliary limbs or severe angulation of the biliary limb due to the loop formation in the remnant stomach or at the gastroenteral and/or jejunojejunal anastomoses [27].

The primary limitations of this study include its retrospective and single-center nature, where AE capture is particularly limited by medical record chart review. While the study population represents one of the largest groups reported for altered anatomy ERCP, it nevertheless represents procedures performed only by endoscopists highly experienced in complex ERCP, limiting its generalizability. Furthermore, the type of endoscope utilized was left to the discretion of the endoscopist, introducing sample bias into the study. Use of the DBE system, for example, often followed failed intubation with an SBE, skewing the study population treated with DBE to have more difficult anatomy to navigate. This is further seen in the limited number of repeat ERCP attempts after initial enteroscopy failure, as we did not protocolize when to re-attempt ERCP. Prospective studies are thus still needed to compare endoscope types for each type of altered anatomy, and our results simply reflect our own clinical practice with our suggestions for each anatomy type depicted in Figure 2. Further, the lack of comparator groups (i.e., interventional radiology-assisted, laparoscopy-assisted) in this study limits the applicability of our results.

With regard to future directions, though the manual rotational enteroscope is no longer commercially available, the first prospective study involving a novel motorized rotational or ‘spiral’ enteroscopy system (Olympus) reported a remarkable 10% rate of intubating the cecum in antegrade examinations, which implies potential utility to facilitate ERCP in particularly long afferent limbs [29]. A recent multicenter study from Japan examined the passive bending colonoscope (PCF PQ260L, Olympus), which allows for smooth advancement of the endoscope without forming a sharp angle, reported a high enteroscopy success rate (91.4%) in a shorter amount of time (10 min) compared to balloon enteroscopy-assisted ERCP [30]. Nevertheless, as one of the primary limitations of forward-viewing endoscopes remains the lack of an elevator, an altered anatomy-specific ERCP device-assisted enteroscope with an elevator would be a natural innovation to current endoscope design, and further investigation into this technology is greatly awaited. Furthermore, devices such as sphincterotomes that allow for easy angulation and rotation may greatly facilitate both cannulation and sphincterotomy in altered anatomy cases with altered angles of approach for successful ductal access [31].

In summary, this study describes a single center’s experience with performing ERCP in surgically altered upper gastrointestinal anatomy using a variety of endoscopes. While caution is warranted in generalizing our results as all ERCPs were performed by expert endoscopists, we demonstrate the relative safety and efficacy of performing ERCP in these challenging anatomy types, primarily with forward-viewing endoscopes. Given its efficacy and its low adverse event rate compared to IR, EUS-guided, and surgical techniques, we believe ERCP should still be offered as a first-line approach for patients with surgically altered anatomy.

## Figures and Tables

**Figure 1 medsci-13-00018-f001:**
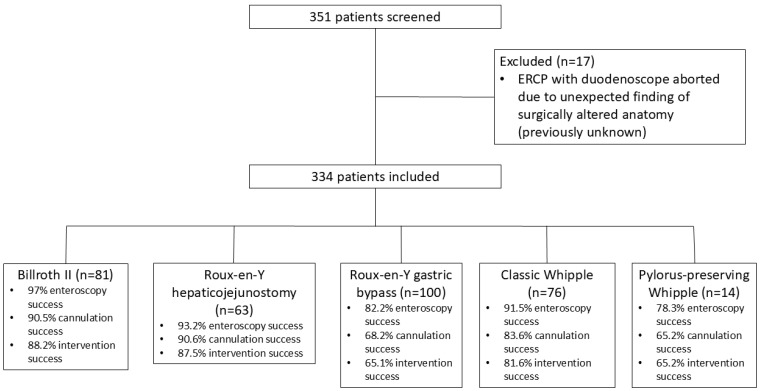
Flowchart of patients.

**Figure 2 medsci-13-00018-f002:**
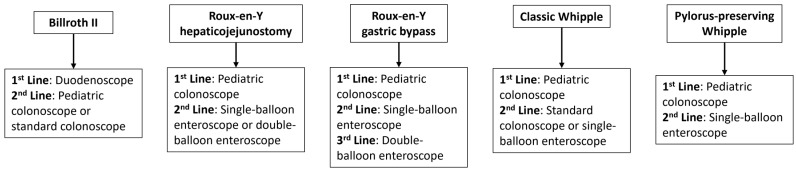
Our approach to each anatomy type based on success rates from this study.

**Table 1 medsci-13-00018-t001:** Demographics and procedural data (334 patients, 665 ERCPs).

Variable	Mean (SD) or n (%)
Age	56.7 (15.5)
Female Sex	167 (50%)
Anatomy Type	
- Billroth II	81 (24.3%)
- Roux-en-Y hepaticojejunostomy	63 (18.9%)
- Roux-en-Y gastric bypass	100 (29.9%)
- Classic pancreaticoduodenectomy	76 (22.8%)
- Pylorus-preserving pancreaticoduodenectomy	14 (4.2%)
Number of ERCPs/patient	1.99 (1.81)
Indications	
- Abdominal pain	156 (22.9%)
- Cholangitis	121 (17.7%)
- Abnormal liver function tests	119 (17.4%)
- Abnormal imaging	81 (11.9%)
- Jaundice	70 (10.3%)
- Anastomotic stricture	44 (6.5%)
Duct of interest	
- Biliary	309 (92.5%)
- Pancreatic	25 (7.5%)
Endoscope Type	
- Duodenoscope	141 (21.2%)
- Pediatric colonoscope	214 (32.2%)
- Adult colonoscope	25 (3.8%)
- Rotational enteroscope	61 (9.2%)
- Single-balloon enteroscope	210 (31.6%)
- Double-balloon enteroscope	14 (2.1%)
Intervention Performed	
- Sphincterotomy	195 (29.3%)
- Dilation	404 (60.8%)
- Stenting	254 (38.2%)
Adverse Events	34 (5.1%)
- Post-ERCP pancreatitis	8 (1.2%)
- Perforation	4 (0.6%)
- Abdominal pain requiring overnight observation	13 (1.9%)

**Table 2 medsci-13-00018-t002:** Success rates by anatomy type.

Anatomy	Enteroscopy Success	Cannulation Success	Intervention Success
Billroth II (81 patients, 169 ERCPs)	164 (97.0%)	153 (90.5%)	149 (88.2%)
Roux-en-Y hepaticojejunostomy (63 patients, 192 ERCPs)	179 (93.2%)	174 (90.6%)	168 (87.5%)
Roux-en-Y gastric bypass (100 patients, 129 ERCPs)	106 (82.2%)	88 (68.2%)	84 (65.1%)
Classic pancreaticoduodenectomy (76 patients, 152 ERCPs)	139 (91.5%)	127 (83.6%)	124 (81.6%)
Pylorus-preserving pancreaticoduodenectomy (14 patients, 23 ERCPs)	18 (78.3%)	15 (65.2%)	15 (65.2%)

**Table 3 medsci-13-00018-t003:** Success rates by endoscopy type.

Endoscope Type	Enteroscopy Success	Cannulation Success	Intervention Success
Duodenoscope	140 (99.3%)	136 (96.5%)	133 (94.3%)
Pediatric Colonoscope	186 (86.9%)	163 (76.2%)	159 (74.3%)
Adult Colonoscope	24 (96%)	24 (96%)	23 (92%)
Rotational Enteroscope	60 (93.4%)	52 (85.3%)	51 (83.6%)
Single-Balloon Enteroscope	187 (89.1%)	174 (82.9%)	167 (79.5%)
Double-Balloon Enteroscope	9 (64.3%)	8 (57.1%)	7 (50%)

**Table 4 medsci-13-00018-t004:** The success rate for Roux-en-Y hepaticojejunostomy anatomy by endoscope utilized.

Anatomy	Enteroscopy Success	Cannulation Success	Intervention Success
Roux-en-Y Hepaticojejunostomy (63 Patients, 192 ERCPs)			
Pediatric colonoscope (n = 13)	13 (100%)	11 (84.6%)	11 (84.6%)
Adult colonoscope (n = 6)	6 (100%)	6 (100%)	6 (100%)
Rotational enteroscope (n = 26)	25 (96.2%)	25 (96.2%)	24 (92.3%)
Single-balloon enteroscope (n = 138)	130 (94.2%)	128 (92.8%)	123 (89.1%)
Double-balloon enteroscope (n = 9)	5 (55.6%)	4 (44.9%)	4 (44.9%)

**Table 5 medsci-13-00018-t005:** The success rate for Roux-en-Y gastric bypass anatomy by endoscope utilized.

Anatomy	Enteroscopy Success	Cannulation Success	Intervention Success
Roux-en-Y Gastric Bypass (100 Patients, 129 ERCPs)			
Pediatric colonoscope (n = 41)	29 (70.7%)	26 (63.4%)	26 (63.4%)
Adult colonoscope (n = 3)	2 (66.7%)	2 (66.7%)	1 (33.3%)
Rotational enteroscope (n = 28)	28 (100%)	21 (75%)	21 (75%)
Single-balloon enteroscope (n = 54)	45 (83.3%)	37 (68.5%)	35 (64.8%)
Double-balloon enteroscope (n = 3)	2 (66.7%)	2 (66.7%)	1 (33.3%)

## Data Availability

The data presented in this study are available on request from the corresponding author due to privacy reasons.

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
