# Peer review of "The Success and Safety of Endoscopic Retrograde Cholangiopancreatography in Surgically Altered Gastrointestinal Anatomy"

_medsci, 2025, doi:10.3390/medsci13010018_

Round 1
Reviewer 1 Report
Comments and Suggestions for Authors
1.ABSTRACT:
The abbreviation ERCP is not explained! There are other abbreviations in the abstract that are not detailed! Are the patients adults or children?
2.ARTICLE:
This is a descriptive study over a period of 17 years! The article is well written, I only have a few minor observations:
- What is new in your article? Please include the scientific novelty you brought through this retrospective study in the title, the conclusions in the abstract, and the final conclusions of the article;
- Please cut out this paragraph:"Further prospective comparative studies are needed to effectively compare different modalities of pancreaticobiliary access but our data suggests that ERCP should be considered as a 1st-line therapeutic option for patients with surgically altered anatomy. " In the conclusions, state the particularity of your study, even though it is a single-center retrospective study.
Author Response
1.ABSTRACT:
The abbreviation ERCP is not explained! There are other abbreviations in the abstract that are not detailed! Are the patients adults or children?
We thank the Reviewer for noting this. We have clarified all the abbreviations in the abstract and clarified that these are all adults.
2.ARTICLE:
This is a descriptive study over a period of 17 years! The article is well written, I only have a few minor observations:
- What is new in your article? Please include the scientific novelty you brought through this retrospective study in the title, the conclusions in the abstract, and the final conclusions of the article
We appreciate the Reviewer’s comment and have now emphasized how the primary novelty is the inclusion of many different surgically altered anatomy types with specific success rates for a wide variety of endoscopes, with our findings demonstrating that altered anatomy ERCP with forward-viewing endoscopes is effective enough to warrant its use as the first-line modality.
- Please cut out this paragraph:"Further prospective comparative studies are needed to effectively compare different modalities of pancreaticobiliary access but our data suggests that ERCP should be considered as a 1st-line therapeutic option for patients with surgically altered anatomy. " In the conclusions, state the particularity of your study, even though it is a single-center retrospective study.
In line with the previous comment, we have removed this section and highlighted the value of our study in reinforcing ERCP as the first-line approach for pancreaticobiliary pathology in patients with surgically altered anatomy.
Reviewer 2 Report
Comments and Suggestions for Authors
The paper describes various methods to perform ERCP in surgically altered anatomy. In this day and age of increased metabolic surgery, certainly, many patients have altered anatomy.
It is refreshing to see that this paper highlights that endoscopy approaches can be performed first before embarking on more invasive procedures such as IR guided or surgically guided. The success rates ranging between 60-100% should be applauded albeit with a disclaimer statement that these procedures were performed by master endoscopists.
It would be nice to have a recommendation from the authors based on their experience as in the choice of procedure and endoscope to use to manage different altered anatomy. Maybe this could be included in the conclusion as a flow chart?
Otherwise, a well presented paper with conclusions reflecting the results from the data.
Author Response
The paper describes various methods to perform ERCP in surgically altered anatomy. In this day and age of increased metabolic surgery, certainly, many patients have altered anatomy.
It is refreshing to see that this paper highlights that endoscopy approaches can be performed first before embarking on more invasive procedures such as IR guided or surgically guided. The success rates ranging between 60-100% should be applauded albeit with a disclaimer statement that these procedures were performed by master endoscopists.
We thank the Reviewer for noting this and we have revised our paper to reflect that highly experienced endoscopists performed this study.
It would be nice to have a recommendation from the authors based on their experience as in the choice of procedure and endoscope to use to manage different altered anatomy. Maybe this could be included in the conclusion as a flow chart?
We agree with the Reviewer for this suggestion and have included a flow chart (Figure 2) with recommendations for each type of anatomy.
Otherwise, a well presented paper with conclusions reflecting the results from the data.
We thank the Reviewer for their favorable comments.